# Rivaroxaban Monotherapy in Patients with Pulmonary Embolism: Off-Label vs. Labeled Therapy

**DOI:** 10.3390/life12081128

**Published:** 2022-07-27

**Authors:** Pierpaolo Di Micco, Vladimir Rosa Salazar, Carmen Fernandez Capitan, Francesco Dentali, Covadonga Gomez Cuervo, Jose Luis Fernandez Torres, Jose Antonio Porras, Angeles Fidalgo, Elvira Grandone, Manuel Lopez Meseguer, Manuel Monreal

**Affiliations:** 1UOC Medicina y Urgencia—ASL NAPOLI 2 Nord Ospedale Anna Rizzoli di Lacco Ameno, 34110 Naples, Italy; 2Department of Internal Medicine, Hospital Universitario Virgen de Arrixaca, 30627 Murcia, Spain; vladi_medico@yahoo.es; 3Department of Internal Medicine, Hospital Universitario La Paz, 28015 Madrid, Spain; mfcapitan@salud.madrid.org; 4Department of Medicine and Surgery, Insubria University, 21100 Varese, Italy; francesco.dentali@asst-settelaghi.it; 5Department of Internal Medicine, Hospital Universitario 12 de Octubre, 28015 Madrid, Spain; cova.gomez.cuervo@gmail.com; 6Department of Internal Medicine, Complejo Hospitalario de Jaén, 23160 Jaén, Spain; jlfernandezreyes@gmail.com; 7Department of Internal Medicine, Hospital Universitario Joan XXIII de Tarragona, 43003 Tarragona, Spain; aporras.hj23.ics@gencat.cat; 8Department of Internal Medicine, Hospital Universitario de Salamanca, 37004 Salamanca, Spain; angelesfidalgo@gmail.com; 9Atherosclerosis and Thrombosis Unit, Casa Sollievo Della Sofferenza, 71121 Foggia, Italy; e.grandone@operapadrepio.it; 10Medical and Surgical Sciences Dept., University of Foggia, 71121 Foggia, Italy; 11Ob/Gyn Dept., First Moscow State Medical University, 101000 Moscow, Russia; 12Department of Pneumonology, Hospital Universitario Vall d’Hebron, 08016 Barcelona, Spain; manuelopez@vhebron.net; 13Chair for the Study of Thromboembolic Disease, Faculty of Health Sciences, UCAM—Universidad Católica San Antonio de Murcia, 30627 Murcia, Spain

**Keywords:** oral anticoagulants, DOACs, rivaroxaban, venpus thromboembolism, RIETE

## Abstract

Background: The use of rivaroxaban in clinical practice often deviates from manufacturer prescribing information. No studies have demonstrated an association between this practice and improved outcomes. Methods: We used the RIETE registry to assess the clinical characteristics of patients with pulmonary embolism (PE) who received off-label rivaroxaban, and to compare their 3-month outcomes with those receiving the labeled therapy. The patients were classified into four subgroups: (1) labeled therapy; (2) delayed start; (3) low doses and (4) both conditions. Results: From May 2013 to May 2022, 2490 patients with PE received rivaroxaban: labeled therapy—1485 (58.6%); delayed start—808 (32.5%); low doses—143 (5.7%); both conditions—54 (2.2%). Patients with a delayed start were more likely to present with syncope, hypotension, raised troponin levels and more severe abnormalities on the echocardiogram than those on labeled therapy. Patients receiving low doses were most likely to have cancer, recent bleeding, anemia, thrombocytopenia or renal insufficiency. During the first 3 months, 3 patients developed PE recurrence, 4 had deep-vein thrombosis, 11 had major bleeding and 16 died. The rates of major bleeding (11 vs. 0; *p* < 0.001) or death (15 vs. 1; OR: 22.5; 95% CI: 2.97–170.5) were higher in patients receiving off-label rivaroxaban than in those on labeled therapy, with no differences in VTE recurrence (OR: 1.11; 95% CI: 0.25–6.57). Conclusions: In patients with severe PE, the start of rivaroxaban administration was often delayed. In those at increased risk for bleeding, it was often prescribed at low doses. Both subgroups had a worse outcome than those on labeled rivaroxaban.

## 1. Introduction

Pulmonary embolism (PE) is a potentially life-threatening disease, spanning a wide spectrum of clinical outcomes [1]. The current guidelines on antithrombotic therapy recommend the use of direct oral anticoagulants (DOACs), such as rivaroxaban or apixaban, as initial and long-term therapy for patients with acute PE and clinical characteristics similar to those treated in phase III trials [2]. The use of DOACs has been associated with a similar degree of antithrombotic effect and a significant reduction in risk for major bleeding. Rivaroxaban was first approved for the treatment of PE in 2011. Since its approval, a large body of evidence for the safety and effectiveness of rivaroxaban in daily clinical practice has been collected. Real-world evidence is associated with higher external validity compared with randomized clinical trials; the latter, often represent a controlled environment that provides high internal validity but can fail to reflect the true nature of clinical practice due to tight exclusion criteria. Thus, this analysis is aimed at assessing the real-world effectiveness and safety of rivaroxaban in real-world patients with acute PE.

In the EINSTEIN–PE trial, rivaroxaban was used at a dose of 30 mg twice daily for 21 days, followed by 20 mg once daily for at least 3 months [3]. Patients were ineligible if they had received unfractionated heparin (UFH), low-molecular-weight heparin (LMWH) or fondaparinux for over 48 h, or if they had creatinine clearance (CrCl) levels < 30 mL/min, a high risk of bleeding, or a life expectancy of <3 months. However, prior studies performed in patients with venous thromboembolism (VTE) [4,5,6,7,8,9] and atrial fibrillation [10,11,12,13,14] found that in clinical practice, the use of DOACs deviates from manufacturer prescribing information by rates of 17% to 50%. Although prescribers may have valid reasons for using off-label DOACs, no studies have yet demonstrated an association between this practice and improved outcomes. Some studies have documented an association between off-label use and increased risk of adverse events, [12,13] but others have not [9,15,16,17].

The RIETE (Registro Informatizado Enfermedad TromboEmbólica) database is an ongoing, multicenter, observational registry enrolling consecutive patients with objectively confirmed acute venous thromboembolism (VTE). It started in Spain in 2001, with the aim of providing information on the natural history of patients with VTE, and subsequently expanded to hospitals in other countries (ClinicalTrials.gov identifier: NCT02832245). The aim of the current study was to assess the clinical characteristics of PE patients who received off-label rivaroxaban, and to compare their 3-month outcomes with those receiving the labeled therapy.

## 2. Methods

### 2.1. Data Source

RIETE included consecutive patients with acute symptomatic deep-vein thrombosis (DVT) or PE, confirmed using objective tests (pulmonary angiography, ventilation–perfusion lung scan, or helical computed tomography [CT] scan for suspected PE, and compression ultrasonography for suspected DVT). The design and conduct of the RIETE registry have been described previously [18]. All patients (or their relatives) provided written or oral informed consent for participation in the registry, in accordance with the local ethics committee requirements. Patients were excluded if they were currently participating in a blind therapeutic clinical trial.

### 2.2. Study Design

For the purposes of this study, we selected only patients presenting with acute symptomatic PE (with or without concomitant DVT) receiving rivaroxaban from May 2013 to May 2022 (rivaroxaban was licensed to the market for VTE treatment in European countries in October 2012). Patients with incidental PE were not included. The primary outcome was the proportion of patients receiving off-label vs. labeled rivaroxaban. Rivaroxaban therapy was classified as off-label (non-concordant with manufacturer labeling) when they were started beyond the second day after the index PE, and/or when lower-than-recommended doses were used (30 mg daily for the first 21 days, followed by 20 mg daily for at least 3 months). Patients were classified into four subgroups: (1) labeled therapy; (2) delayed start (when rivaroxaban was started beyond the second day); (3) low doses (when rivaroxaban was prescribed at lower-than-recommended doses, either initially or long-term); and (4) both conditions. In patients who started rivaroxaban therapy beyond the second day after the index PE, the drugs, doses and duration of the initial therapy were reported. Secondary outcomes included factors associated with off-label use of rivaroxaban and the incidence rates of symptomatic VTE recurrence, major bleeding, or death occurring within the first 90 days of therapy. Patients who developed VTE-related outcomes during initial therapy with other drugs were excluded for the current analysis.

All episodes of clinically suspected VTE recurrence were investigated via repeat compression ultrasonography, helical CT pulmonary, ventilation–perfusion lung scintigraphy, angiography, or pulmonary angiography. Recurrent VTE was defined as a DVT in a new segment, a DVT of 4 mm larger in diameter when compared with the prior venous ultrasound, a new ventilation–perfusion mismatch in a repeat lung scan, or a new intraluminal filling defect on a CT scan. Bleeding complications were classified as major if they were overt and required a transfusion of two units of blood or more; if they were retroperitoneal, spinal, or intracranial; or if they were fatal. Fatal PE, in the absence of autopsy, was defined as any death appearing within 10 days after symptomatic PE diagnosis, in the absence of any alternative cause of death. Fatal bleeding was considered any death appearing within 10 days after a major bleed, in the absence of any alternative cause of death.

### 2.3. Study Variables

The following variables were recorded in RIETE: patient’s characteristics; PE signs and symptoms at baseline; clinical status, including any coexisting or underlying conditions such as chronic heart or lung disease, recent (<30 days before) major bleeding, anemia or renal insufficiency; concomitant disorders; risk factors for VTE; concomitant drugs; the treatment received upon PE diagnosis (drugs, doses and duration); and the outcomes during at least the first 90 days. The index PEs were categorized as subsegmental only, segmental (with or without subsegmental arteries) or central PE (lobar or pulmonary arteries, with or without segmental or subsegmental arteries), based on the largest arteries involved in the CT scan.

Immobilized patients were defined as non-surgical patients who had been immobilized (i.e., total bed rest with or without bathroom privileges) for ≥4 days in the 2-month period prior to PE diagnosis. Surgical patients were defined as those who had undergone an operation in the 2 months prior to the index PE. Active cancer was defined as newly diagnosed cancer (<3 months before) or cancer while receiving anti-neoplastic treatment of any type (i.e., chemotherapy, radiotherapy, surgery, hormonal therapy, support therapy or combined therapies). Recent major bleeding was defined as any major bleeding occurring within the first 30 days prior to the index PE. Anemia was defined as hemoglobin levels <12 g/dL in women, or <13 g/dL in men. CrCl levels at baseline were calculated using the Cockcroft and Gault formula.

### 2.4. Treatment and Follow-Up

There was no standardized treatment for patients in RIETE. Patients were managed according to the recommendations from their attending clinicians and the participating hospitals. All (100%) patients in the RIETE registry underwent follow-ups for 90 days or until death. Longer follow-ups were available for most patients. During each visit, the occurrence of VTE recurrence or major bleeding was assessed. For patients who developed VTE recurrence or major bleeding during the first 90 days, a minimum follow-up of 30 days (or until death) occurred for all study participants after the event, irrespective of when it happened.

### 2.5. Statistical Analysis

Categorical variables were reported as frequency counts (percentages) and compared using the chi-square test (two-sided) and Fisher’s exact test (two-sided). Continuous variables were reported as the mean and standard deviation (or median with inter-quartile range, if not normally distributed), and compared using a Student’s t-test or Mann–Whitney test, as appropriate. Odds ratios (OR) and the corresponding 95% confidence intervals (CIs) were calculated. Statistical analyses were conducted using IBM SPSS Statistics (version 25).

## 3. Results

From May 2013 to May 2022, there were 28,317 patients with acute symptomatic PE in the RIETE registry. Of these, 2490 (8.8%) received rivaroxaban, 6866 (24.2%) received other DOACs and 18,961 (67.0%) did not receive DOACs. Patients on rivaroxaban were more likely to be men and significantly younger, and were less likely to have cancer, recent major bleeding, anemia, thrombocytopenia or renal insufficiency than those in the other two subgroups not receiving rivaroxaban (Table 1).

Among the 2490 patients treated with rivaroxaban, 1485 (58.6%) received labeled therapy; 808 (32.5%) had a delayed start (median: 5 days; inter-quartile range: 4–7 days); 143 (5.7%) received low doses, and 54 (2.2%) had both conditions. Among the 862 patients who started rivaroxaban beyond the second day, 89% had initially received LMWH, 9.7% had received UFH and 1.3% had received fondaparinux. Among those receiving low doses, 21.7% received 20 mg daily initially, and 34.3% received 15 mg (Table 2). As for long-term therapy, 41.3% of patients received 15 mg daily, 5.6% had 10 mg, and 3.5% received 5 mg daily. The proportion of PE patients who received rivaroxaban at home (not requiring hospital admission) was much higher in patients receiving labeled therapy than in those on off-label therapy: 24.0% vs. 6.9%, respectively (Table 2).

Patients with a delayed start were older and less likely to have immobility than those on labeled therapy, but were more likely to have had recent surgery (Table 3). They also were more likely to have signs of severe PE (hypotension, tachycardia) or raised troponin levels or echocardiographic findings (raised pulmonary artery pressure levels), and a higher burden of the PE on a CT scan, than patients on labeled therapy. Patients receiving rivaroxaban at low doses also were older and more likely to have cancer, recent major bleeding, anemia, thrombocytopenia or renal insufficiency than those receiving labeled therapy. However, there were no differences in the proportion of patients with hypotension or tachycardia, nor in the burden of PE on the CT scan (Table 3).

During the first 3 months of rivaroxaban therapy, 3 patients developed PE recurrence, 4 had DVT, 11 suffered major bleeding and 16 died (one from fatal PE and one from fatal bleeding). The rates of major bleeding (11 vs. 0; *p* < 0.001) or death (15 vs. 1; OR: 22.5; 95% CI: 2.97–170.5) were significantly higher in patients receiving off-label rivaroxaban than in those on labeled therapy (Table 4). There were no differences in the rate of VTE recurrence (OR: 1.11; 95% CI: 0.25–6.57).

## 4. Discussion

Our findings, obtained from a large cohort of patients with acute PE treated with rivaroxaban, reveal that one in every three (32.5%) had a delayed start, one in every seventeen (5.7%) received lower-than-recommended doses and one in every forty-six (2.2%) met both conditions. This is important since all these therapeutic decisions were associated with worse outcomes. Worse outcomes occurred not only patients receiving low doses, but also in those on recommended doses of rivaroxaban who started after several days of heparin. Compared to those who received labeled therapy, the proportion of patients developing major bleeding or death within the first 3 months was much higher in patients with a delayed start and in those receiving low doses. There were no differences in the rate of VTE recurrence. Moreover, one in every four patients (24%) on labeled therapy was treated at home (not requiring hospital admission), compared to only a minority (6.9%) of those receiving off-label therapy.

Patients with a delayed start were more likely to have severe PE at baseline: they were more likely to have hypotension and raised troponin levels, and had more proximal PEs on the CT scan and more severe abnormalities on the echocardiogram. All of them had received a short course (median, 5 days) of LMWH or UFH before switching to rivaroxaban. Of course, we are unaware of how many additional patients starting with LMWH would also have switched to rivaroxaban if they had not suffered an adverse event during these few days. Thus, our data cannot be used to compare early vs. delayed starts of rivaroxaban. The higher mortality rate is likely to be explained by the higher severity of the index PE. The (non-significantly) higher rate of major bleeding deserves an explanation, with more patients and a different design.

Patients receiving low doses of rivaroxaban were more likely to have recent major bleeding, anemia, thrombocytopenia or renal insufficiency. It is conceivable that their attending doctors were concerned about their risk for bleeding, and this may be the reason why they prescribed lower-than-recommended doses of rivaroxaban. Not unexpectedly, they had the highest rate of major bleeding, and this is likely to, be the consequence of their clinical condition rather than a side effect of rivaroxaban therapy. Interestingly, however, no patients on low-dose rivaroxaban developed VTE recurrence. Thus, our findings may help to reassure patients that low-dose rivaroxaban does not seem to be associated with an increased risk for VTE recurrence.

In the literature, the association between the off-label use of DOACs and its outcomes remains controversial. In a review of 75 studies of patients with atrial fibrillation, 25–50% of them received off-label doses of DOACs [13]. Overdosing was associated with increased mortality and worse bleeding events, while underdosing was associated with a nearly 5-fold increased risk of stroke. However, in a cohort study on 8425 patients with atrial fibrillation, 39% of the patients were treated with off-label dose-reduced DOACs, and they had a reduced effectiveness without a safety benefit [14]. Other studies also provided controversial findings [9,15,16,17].

Our findings have to be interpreted in view of some limitations. First, since RIETE is an observational registry, patients were not treated according to a standardized therapeutic scheme. PE treatment varied according to local practices and the practices’ evolution over time. In this regard, the majority of patients with cancer were treated with LMWH for initial and long-term therapy, with few of them receiving rivaroxaban or other DOACs. Second, one in every three patients in our cohort (34.6%) had received LMWH, UFH or fondaparinux for >48 h before switching to rivaroxaban. However, those who had developed VTE-related outcomes (VTE recurrence or bleeding) during initial therapy with other drugs were excluded from participating in the current analyses. Third, we were not able to assess whether the higher rates of major bleeding and death in patients with off-label rivaroxaban therapy were influenced by the baseline characteristics of the patients. Finally, there is no centralized quality-control in the RIETE registry regarding the accuracy of the outcomes. For technical reasons, it is not possible to review all the imaging tests from over 200 participating centers in 26 countries. Our study also has definite strengths; in particular, the large sample size allowed for the precision of our results and the objective documentation of both incident and recurrent PE.

## 5. Conclusions

In conclusion, we found two main subgroups of PE patients receiving off-label rivaroxaban. In patients presenting with severe signs and symptoms of PE, the start of the drug was often delayed. In those perceived to be at increased risk for bleeding, rivaroxaban was often prescribed at lower-than-recommended doses. Both subgroups had worse outcomes than those receiving labeled rivaroxaban.

## Figures and Tables

**Table 1 life-12-01128-t001:** Patients with pulmonary embolism, according to use of rivaroxaban, other DOACs and other drugs.

	Rivaroxaban	Other DOACs	Other Drugs
Patients, N	2490	6866	18,961
Demographics			
Male gender	1302 (52.3%)	3417 (49.8%) *	9268 (48.9%) ^†^
Age (mean years ± SD)	60 ± 17	65 ± 17 ^‡^	67 ± 16 ^‡^
Age < 50 years	668 (26.8%)	1414 (20.6%) ^‡^	2967 (15.6%) ^‡^
Age > 80 years	278 (11.2%)	1422 (20.7%) ^‡^	4453 (23.5%) ^‡^
Body weight (mean kg ± SD)	80 ± 17	79 ± 17 ^†^	78 ± 18 ^‡^
Risk factors for PE,			
Active cancer	135 (5.4%)	588 (8.6%) ^‡^	3830 (20.2%) ^‡^
Recent surgery	251 (10.1%)	633 (9.2%)	1933 (10.2%)
Recent immobility ≥ 4 days	530 (21.3%)	1610 (23.4%) *	4235 (22.3%)
Pregnancy or postpartum	7 (0.3%)	54 (0.8%) ^†^	162 (0.8%) ^†^
Estrogen use	226 (9.1%)	454 (6.6%) ^‡^	1042 (5.5%) ^‡^
Unprovoked	1436 (57.7%)	3877 (56.5%)	9403 (49.6%) ^‡^
Comorbidities,			
Recent major bleeding	29 (1.2%)	164 (2.4%) ^‡^	539 (2.8%) ^‡^
Anemia	542 (21.8%)	1774 (25.8%) ^‡^	6721 (35.4%) ^‡^
Platelet count < 100,000/µL	18 (0.7%)	85 (1.2%) *	540 (2.8%) ^‡^
CrCl levels 30–60 mL/min	445 (17.9%)	1465 (21.3%) ^‡^	5104 (26.9%) ^‡^
CrCl levels < 30 mL/min	27 (1.1%)	215 (3.1%) ^‡^	1050 (5.5%) ^‡^

Comparisons between patients receiving other DOACs or other anticoagulant drugs and those receiving rivaroxaban: * *p* < 0.05; ^†^
*p* < 0.01; ^‡^
*p* < 0.001. Abbreviations: DOACs—direct oral anticoagulants; SD—standard deviation; PE—pulmonary embolism; CrCl—creatinine clearance.

**Table 2 life-12-01128-t002:** Therapeutic strategies among PE patients receiving rivaroxaban.

	Labeled Therapy	Off-Label Therapy		
		Delayed Start	Low Doses	Both
Patients, N	1485	808	143	54
Rivaroxaban initially				
30 mg daily	1485 (100%)	808 (100%)	63 (44.1%) ^‡^	26 (48.1) ^‡^
20 mg daily	0	0	31 (21.7%) ^‡^	18 (33.3%)
15 mg daily	0	0	49 (34.3%) ^‡^	10 (18.5%)
Median days (IQR) to start	0 (0–1)	5 (4–7)	0 (0–1)	4 (3–6)
Rivaroxaban long-term				
20 mg daily	1482 (99.8%)	805 (99.6%)	68 (47.6%)	21 (38.9%) ^‡^
15 mg daily	0	0	59 (41.3%)	26 (48.1%) ^‡^
10 mg daily	0	0	8 (5.6%)	5 (9.3%) *
5 mg daily	0	0	5 (3.5%)	0
Median days (IQR) to start	22 (21–23)	25 (22–28)	21 (11–23)	15 (10–25)
Treatment				
Home therapy	355 (24.0%)	46 (5.7%) ^‡^	23 (16.1%) *	0 ^†^
Length of hospital stay				
Median days (IQR)	6 (3–8)	6 (4–9)	7 (5–10) ^‡^	8 (6–11) ^‡^

Comparisons between patients receiving labeled therapy and those receiving off-label therapy: * *p* < 0.05; ^†^
*p* < 0.01; ^‡^
*p* < 0.001. Abbreviations: PE—pulmonary embolism; IQR—inter-quartile range.

**Table 3 life-12-01128-t003:** Signs, symptoms, severity of the index PE and length of hospital stay, according to the treatment regimen.

	Labeled Therapy	Off-Label Therapy		
		Delayed Start	Low Doses	Both
Patients, N	1485	808	143	54
Demographics				
Male gender	787 (53.0%)	429 (53.1%)	61 (42.7%) *	25 (46.3%)
Age (mean years ± SD)	59 ± 17	62 ± 17 ^‡^	64 ± 19 ^‡^	72 ± 14 ^‡^
Age > 80 years	141 (9.5%)	88 (10.9%)	32 (22.4%) ^‡^	17 (31.5%) ^‡^
Body weight (mean kg ± SD)	80.5 ± 16.9	81.1 ± 17.3	77.4 ± 17.9	79.2 ± 15.7
Risk factors for PE				
Active cancer	85 (4.2%)	30 (3.7%)	18 (12.6%) ^†^	2 (3.7%)
Recent surgery	126 (8.5%)	104 (12.9%) ^†^	12 (8.4%)	9 (16.7%) *
Recent immobility ≥ 4 days	358 (24.1%)	146 (18.1%) ^‡^	19 (13.3%) ^†^	7 (12.9%)
Pregnancy or postpartum	3 (0.2%)	4 (0.5%)	0	0
Estrogen use	152 (10.2%)	60 (7.4%) *	14 (9.8%)	0
Comorbidities				
Recent major bleeding	9 (0.6%)	12 (1.5%) *	8 (5.6%) ^‡^	0
Anemia	298 (20.1%)	177 (21.9%)	48 (33.6%) ^‡^	19 (35.2%) *
Platelet count < 100,000/µL	5 (0.3%)	7 (0.9%)	6 (4.2%) ^‡^	0
CrCl levels 30–60 mL/min	237 (15.9%)	154 (19.1%)	37 (25.9%) ^†^	18 (33.3%) ^†^
CrCl levels < 30 mL/min	8 (0.5%)	6 (0.7%)	8 (5.6%) ^‡^	4 (7.4%) ^‡^
PE symptoms				
Dyspnea	1167 (78.6%)	644 (79.7%)	109 (76.2%)	45 (83.3%)
Chest pain	766 (51.6%)	390 (48.3%)	64 (44.8%)	24 (44.4%)
Hemoptysis	66 (4.4%)	41 (5.1%)	12 (8.4%)	2 (3.7%)
Syncope	109 (7.3%)	106 (13.1%) ^‡^	7 (4.9%)	6 (11.1%)
PE signs				
SBP levels < 90 mm Hg	14 (0.9%)	20 (2.5%) ^†^	2 (1.4%)	2 (3.7%)
Heart rate > 110 bpm	179 (12.4%)	137 (17.3%) ^†^	15 (11.5%)	3 (5.7%)
Burden of PE on CT scan				
Subsegmental only	101 (7.9%)	47 (6.5%)	11 (9.8%)	4 (8.9%)
Segmental	411 (32.1%)	170 (23.4%) ^‡^	37 (33.0%)	13 (28.9%)
Lobar	467 (36.4%)	201 (27.6%) ^‡^	37 (33.0%)	14 (31.1%)
More proximal	297 (23.2%)	309 (42.5%) ^‡^	27 (24.1%)	14 (31.1%)
Echocardiogram				
RV hypokinesis (N = 1227)	43 (6.4%)	80 (16.8%) ^‡^	6 (13.0%)	1 (3.6%)
PAP levels > 50 mm Hg (N = 739)	38 (8.1%)	66 (29.5%) ^‡^	6 (18.2%)	3 (21.4%)
Blood tests				
Raised troponin levels (N = 1681)	190 (19.0%)	256 (46.4%) ^‡^	31 (32.6%) ^†^	14 (43.8%) ^†^
Treatment				
In hospital	1130 (76.0%)	762 (94.3%) ^‡^	120 (83.9%) *	54 (100%) ^†^
Median days in hospital (IQR)	6 (3–8)	6 (4–9)	7 (5–10) ^‡^	8 (6–11) ^‡^

Comparisons between patients receiving labeled therapy and those receiving off-label therapy: * *p* < 0.05; ^†^
*p* < 0.01; ^‡^
*p* < 0.001. Abbreviations: PE—pulmonary embolism; SBP—systolic blood pressure; bpm—beats per minute; RV—right ventricle; PAP—pulmonary artery pressure; IQR—inter-quartile range.

**Table 4 life-12-01128-t004:** Ninety-day outcomes according to the treatment regimen.

	Labeled Therapy	Off-Label Therapy		
		Delayed Start	Low Doses	Both
Patients, N	1485	808	143	54
90-day outcomes				
Recurrent PE	1 (0.1%)	1 (0.1%)	0	1 (1.9%)
Deep-vein thrombosis	3 (0.2%)	1 (0.1%)	0	0
Major bleeding	0	3 (0.4%)	7 (4.9%) ^‡^	1 (1.9%)
Uterine	0	0	3 (2.1%)	0
Intracranial	0	1 (0.1%)	2 (1.4%)	0
Gastrointestinal	0	0	2 (1.4%)	0
Hematoma	0	0	0	1 (1.9%)
Hemoptysis	0	1 (0.1%)	0	0
Other	0	1 (0.1%)	0	0
All-cause death	1 (0.1%)	5 (0.6%) *	9 (6.3%) ^‡^	1 (1.9%)
Fatal PE	1 (0.1%)	0	0	0
Fatal bleeding	0	1 (0.1%)	0	0

Comparisons between patients receiving labeled therapy and those receiving off-label therapy: * *p* < 0.05; ^‡^
*p* < 0.001. Abbreviations: PE—pulmonary embolism.

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
