# Peer review of "Rivaroxaban Monotherapy in Patients with Pulmonary Embolism: Off-Label vs. Labeled Therapy"

_life, 2022, doi:10.3390/life12081128_

Round 1

Reviewer 1 Report

The manuscript has a valuable practical value, but it needs to be improved before publication.

The chapter "introduction" is uninformative. I recommend adding more information about rivaroxaban (mechanism of action, pharmacokinetics, etc.), the prevalence and pathogenesis of PE, and clinical recommendations for the management of patients with PE.

Author Response

Many thanks. We strongly modified the Introduction Section according to your requirements. The first paragraph of the Introduction section currently reads:

“Pulmonary embolism (PE) is a potentially life-threatening disease, spanning a wide spectrum of clinical outcomes.1 Current guidelines on antithrombotic therapy recommend the use of direct oral anticoagulants (DOACs), such as rivaroxaban or apixaban, as initial- and long-term therapy for patients with acute PE and clinical characteristics similar to those treated in phase III trials.2 The use of DOACs has been associated with a similar degree of antithrombotic effect and a significant reduction of the risk for major bleeding. Rivaroxaban was first approved for the treatment of PE in 2011. Since its approval, a large body of evidence for the safety and effectiveness of rivaroxaban in daily clinical practice has been collected. Real-world evidence is associated with higher external validity compared with randomized clinical trials, which often represent a controlled environment that provides high internal validity but can fail to reflect the true nature of clinical practice due to tight exclusion criteria. Thus, this analysis is aimed at assessing the real-world effectiveness and safety of rivaroxaban in real-world patients with acute PE”.

Reviewer 2 Report

This is a well written manuscript with some limitations that are adequately addressed in the discussion. However, I have some concerns the authors should / may address before publication:

Major points:

Methods, page 3, point 2.5: statistical analysis: was the Student t test also used for non-normally distributed data (continuous variables)? Please state / justify. 

Results, Page 6, Table 4: where the increased numbers of major bleedings in the low dose off label group really found to be non-significantly different from that in the labeled therapy group? Please verify. If yes: Page 7, first sentence: state that these findings were statistically not significant. The authors may also reconsider and/or discuss the general meaningfulness of the "composite outcome".      

Discussion, Page 6, last paragraph and page 7, last paragraph: The fact that all patients with delayed start of rivaroxaban treatment received LMWH/UFH beforehand is first mentioned in the discussion (or concealed in the final conclusions) appears to be misleading. Indeed, this important information should be part of the abstract, described in the results section in detail (e.g. co-medication, Table 3) and also addressed in more detail in the discussion.

Minor points:

Results, Page 4, line 5, typo: "...that those in...."

Results, Page 4, lines 3-4 below Table 1 read:  "...54 (2.2%) had both conditions. All patients with delayed start received labeled doses". These statements appear to be contradictory?! 

Results, Page 4, lines 6-8 read: "The proportion of PE patients that received rivaroxaban at home (not requiring hospital admission) was: 24.0%, 5.7%, 16.1% and 0%, respectively (Table 2)". Notwithstanding reference to Table 2, these numbers should be ascribed to the respective condition also in the text.

Results, Page 5, line 2 below Table 2: "(Table 2)" should read: "(Table 3)".

Results, Page 6, Table 4: Better (visually) distinguish sub- from main categories

Discussion, Page 6, line 7: check the "40-fold" for correctness

Author Response

This is a well written manuscript with some limitations that are adequately addressed in the discussion. However, I have some concerns the authors should / may address before publication:

Major points:

Methods, page 3, point 2.5: statistical analysis: was the Student t test also used for non-normally distributed data (continuous variables)? Please state / justify. 

ANSWER:

Many thanks. It was our fault. We modified the text.

It currently reads:

“Continuous variables were reported as mean and standard deviation (or median with interquartile range, if not normally distributed), and compared using Student t test or Mann-Whitney test, as appropriate”.

Results, Page 6, Table 4: where the increased numbers of major bleedings in the low dose off label group really found to be non-significantly different from that in the labeled therapy group? Please verify. If yes: Page 7, first sentence: state that these findings were statistically not significant. The authors may also reconsider and/or discuss the general meaningfulness of the "composite outcome".      

ANSWER:

Many thanks for your interesting comment.

Following your suggestion, we removed any mention to the composite outcome (in the Methods Section, in the Results Section and in the Discussion), and focused on each of the 3 VTE-related outcomes (VTE recurrences, major bleeding or death) separately.

After this modification, the message appears much more clear.

Certainly, the incidence rate of major bleeding was significantly higher in patients receiving rivaroxaban off label. There was a missing symbol of significance (p <0.001) in Table 4.  We apologize for this.

Discussion, Page 6, last paragraph and page 7, last paragraph: The fact that all patients with delayed start of rivaroxaban treatment received LMWH/UFH beforehand is first mentioned in the discussion (or concealed in the final conclusions) appears to be misleading. Indeed, this important information should be part of the abstract, described in the results section in detail (e.g. co-medication, Table 3) and also addressed in more detail in the discussion.

ANSWER:

Many thanks. We modified the first pararaph in the Discussion Section.

It currently reads:

“Our findings, obtained from a large cohort of patients with acute PE treated with rivaroxaban, reveal that one in every three (32.5%) had delayed start, one in every 17 (5.7%) received lower than recommended doses and one in every 46 (2.2%) met both conditions. This is important since all these therapeutic decisions were associated with worse outcomes. Not only patients receiving low doses: those on recommended doses of rivaroxaban but starting after some days of heparin also had a worse outcome”

Minor points:

Results, Page 4, line 5, typo: "...that those in...."

ANSWER:

Corrected, as suggested. We thank you for this.

Results, Page 4, lines 3-4 below Table 1 read:  "...54 (2.2%) had both conditions. All patients with delayed start received labeled doses". These statements appear to be contradictory?! 

ANSWER:

You are perfectly right. We removed this sentence.

Results, Page 4, lines 6-8 read: "The proportion of PE patients that received rivaroxaban at home (not requiring hospital admission) was: 24.0%, 5.7%, 16.1% and 0%, respectively (Table 2)". Notwithstanding reference to Table 2, these numbers should be ascribed to the respective condition also in the text.

ANSWER:

We agree. We modified the text accordingly.

It currently reads:

“The proportion of PE patients that received rivaroxaban at home (not requiring hospital admission) was much higher in patients receiving labelled therapy than in those on off-label therapy: 24.0% vs. 6.9%, respectively (Table II)“

Results, Page 5, line 2 below Table 2: "(Table 2)" should read: "(Table 3)".

ANSWER:

Many thanks. Corrected, as requested.

Results, Page 6, Table 4: Better (visually) distinguish sub- from main categories

ANSWER:

Many thanks. We modified the text, as explained above.

Discussion, Page 6, line 7: check the "40-fold" for correctness

ANSWER:

We modified the text. It currently reads:

“Compared to those who received labeled therapy, the proportion of patients developing major bleeding or death within the first 3 months was much higher in patients with delayed start or in those receiving low doses. There were no differences in the rate of VTE recurrences”

Round 2

Reviewer 2 Report

As mentioned in the last of the three major points made, I still strongly recommend that the heparin treatment of patients with "delayed start" is also clearly mentioned in the abstract,  the methods section ("study design") and the results (text but also Tables II and III). Otherwise, the term "delayed start" appears to be misleading to the reader. 

Author Response

As mentioned in the last of the three major points made, I still strongly recommend that the heparin treatment of patients with "delayed start" is also clearly mentioned in the abstract,  the methods section ("study design") and the results (text but also Tables II and III). Otherwise, the term "delayed start" appears to be misleading to the reader. 

ANSWER:

You are perfectly right. We apologize.

The reason why we did not include this relevant information into the manuscript is because patients developing VTE recurrences or bleeding during initial therapy with other drugs were not considered in the analyses.

However, we understand that we should have informed about this, and have modified the text of the manuscript according to your suggestion.

We included the following sentence in the Methods Section:

“In patients that started rivaroxaban therapy beyond the second day after the index PE, the drugs, doses and duration of the initial therapy were reported”

And also:

“Patients who developed VTE-related outcomes during initial therapy with other drugs were excluded for the current analysis”

We also included the following sentence in the Results Section:

“Among 862 patients who started rivaroxaban beyond the second day, 89% had initially received LMWH, UFH 9.7%, and fondaparinux 1.3%”

And the following sentence among the study limitations in the Discussion section:

“Second, one in every 3 patients in our cohort (34.6%) had received LMWH, UFH or fondaparinux for >48 hours before switching to rivaroxaban. However, those who had developed VTE-related outcomes (VTE recurrences or bleeding) during initial therapy with other drugs were excluded to participate in the current analyses”